

# Biodiversity of symbiotic microalgae associated with meiofaunal marine acoels in Southern Japan

Siratee Riewluang[1] and Kevin C. Wakeman[2,3]

[1] School of Science, Hokkaido University, Sapporo, Hokkaido, Japan
[2] Institute for the Advancement of Higher Education, Hokkaido University, Sapporo, Hokkaido, Japan
[3] Graduate School of Science, Hokkaido University, Sapporo, Hokkaido, Japan

## ABSTRACT

Acoels in the family Convolutidae are commonly found with microalgal symbionts. Convolutids can host green algal *Tetraselmis* and dinoflagellates within the family Symbiodiniaceae and the genus *Amphidinium*. The diversity of these microalgae has not been well surveyed. In this study, we used PCR and culture techniques to demonstrate the biodiversity of *Tetraselmis* and dinoflagellates in symbiosis with meiofaunal acoels. Here, 66 acoels were collected from seven localities around Okinawa, Ishigaki, and Kochi, Japan. While convolutids were heavily represented in this sampling, some acoels formed a clade outside Convolutidae and are potentially a new family of acoels harboring symbiotic microalgae. From the acoels collected, a total of 32 *Tetraselmis* and 26 Symbiodiniaceae cultures were established. Molecular phylogenies were constructed from cultured material (and from total host DNA) using the 18S rRNA gene (*Tetraselmis*) and 28S rRNA gene (dinoflagellates). The majority of *Tetraselmis* sequences grouped within the *T. astigmatica* clade but strains closely related to *T. convolutae*, *T. marina*, and *T. gracilis* were also observed. This is the first report of *Tetraselmis* species, other than *T. convolutae*, naturally associating with acoels. For dinoflagellates, members of *Cladocopium* and *Miliolidium* were observed, but most Symbiodiniaceae sequences formed clusters within *Symbiodinium*, grouping with *S. natans*, or sister to *S. tridacnidorum*. Several new *Symbiodinium* sequences from this study may represent novel species. This is the first molecular record of *Miliolidium* and *Symbiodinium* from acoels. Microalgal strains from this study will provide a necessary framework for future taxonomic studies and research on symbiotic relationships between acoels and microalgae.

# INTRODUCTION

Acoels (Acoela, Acoelomorpha) are soft-bodied invertebrates that primarily live in marine environments (*Achatz et al., 2013*). Due to their relatively simple body plan, select lineages (*e.g.*, *Symsagittifera roscoffensis*) act as important model organisms in the fields of regenerative research (*Arboleda et al., 2018*) and developmental (neural) sciences (*Bailly et al., 2014*; *Raikova et al., 2004*). Acoels are also of interest in the field of symbiosis because

Corresponding author
Kevin C. Wakeman,
wakeman.k@oia.hokudai.ac.jp

select lineages acquire and maintain photosynthetic microalgae. Within Acoela, the Convolutidae is the only family known to possess microalgal symbionts (*Jondelius & Jondelius, 2020*; *Achatz et al., 2010*). A high diversity of convolutids can be found in subtropical/tropical waters associated with corals (*Barneah et al., 2012*; *Kunihiro et al., 2019*), as epiphytes on macroalgae (*Asai et al., 2022*), or living as microscopic meiofauna between grains of sand (*Thomas, Coates & Tang, 2022*; *Trench & Winsor, 1987*).

The Convolutidae (~100 species) includes 24 genera, 19 of which have images depicting an association with microalgal symbionts (*Tyler et al., 2006–2022*). Hosts typically associate with one symbiont, but up to two symbionts can be found within a single host. For example, *Symsagittifera* and *Convolutriloba* species harbor only *Tetraselmis*, a green algal chlorophyte (*Balzer, 1999*; *Parke & Manton, 1967*), while *Waminoa* are found with two types of dinoflagellates, Symbiodiniaceae and *Amphidinium* (*Barneah et al., 2007*; *Ogunlana et al., 2005*). The genus *Amphiscolops* contain acoels (*e.g.*, *A. oni* and *A. potocani*) that harbor *Tetraselmis* and a dinoflagellate (*Achatz, 2008*; *Asai et al., 2022*). *Kunihiro & Reimer (2018)* is the only study that has surveyed the molecular diversity of microalgal symbionts in acoels. Although they concluded that Symbiodiniaceae within *Waminoa* acoels were different to those found in the acoel's host coral, only a single lineage, *Cladocopium*, was recorded. To our knowledge, no other study has explored the biodiversity of microalgal symbionts of acoels.

*Tetraselmis* (Chlorodendrophyceae, Chlorophyta) is globally distributed (*John, Whitton & Brook, 2011*). They are green unicellular organisms with four equal-length flagella (*Butcher, 1959*). All known *Tetraselmis* (~30 species) can be found free-living, mostly in marine and brackish environments (*Arora, 2017*). Mass cultivations of *Tetraselmis* have been used in applications such as wastewater treatment (*Goswami et al., 2022*), biofuel production (*Teo et al., 2014*), and pharmacology (*Schüler et al., 2020*). Few (if any) of these studies have examined *Tetraselmis* from a biodiversity perspective. Therefore, the taxonomy and systematics of this group remain unresolved. The phylogeny and taxonomy of *Tetraselmis* have mainly been based on pyrenoid or flagella ultrastructure, as their outer appearance can vary (*Butcher, 1959*; *Hori, Norris & Chihara, 1982*, *1983*, *1986*; *Hori & Chihara, 1974*; *Marin, Matzke & Melkonian, 1993*). Morphological-based phylogeny, however, has been questioned after molecular work revealed multiple species were genetically identical (*Arora et al., 2013*; *Hyung et al., 2021*). Although *Tetraselmis* is widespread among acoels (over half of known convolutids contain *Tetraselmis* symbionts), no research has been performed to clarify the diversity of *Tetraselmis* species naturally associating with acoels. To date, only *T. convolutae* from *Symsagittifera roscoffensis* has been identified to the species level (*Parke & Manton, 1967*).

The family Symbiodiniaceae and the genus *Amphidinium* are the only known dinoflagellate symbionts of acoels (*Taylor, 1971*; *Kunihiro et al., 2019*). It is difficult to distinguish between the two types of symbionts, as line drawings from past literature are somewhat ambiguous and refer to both as "zooxanthellae" (*Dorjes, 1968*; *Hyman, 1939*). Symbiodiniaceae are well known as symbionts of corals and can associate with a wide variety of hosts including sponges (*Carlos et al., 1999*), molluscs (*Lee, Jeong & Lajeunesse, 2020*), protists (Foraminifera) (*Pochon et al., 2007*; *Pochon & LaJeunesse, 2021*), and acoels

(*Taylor, 1971*; *Kunihiro & Reimer, 2018*). As for acoel-associated *Amphidinium*, past research focused largely on natural product chemistry, looking at cytological effects of amphidinols and other polyketides extracted from mass cultivations (*Satake, Murata & Yasumoto, 1991*; *Yang et al., 2023*). One of many species used as a source of amphidinols was *Amphidinium gibbosum* which is also found as a symbiont of the acoel *Heterochaerus langerhansi* (*Taylor, 1971*). *Amphidinium gibbosum* is also a symbiont of *Waminoa litus* (*Hikosaka-Katayama et al., 2012*) and presumably *Amphiscolops oni* (*Asai et al., 2022*). Symbiodiniaceae and *Amphidinium* have been reported as symbionts within several genera of convolutids (*Kunihiro et al., 2019*; *Lopes & Silveira, 1994*). Nevertheless, *A. gibbosum* is the only acoel-associated dinoflagellate that has been identified to the species level (*Taylor, 1971*; *Trench & Winsor, 1987*); *Cladocopium* is the only acoel-associated Symbiodiniaceae genus that has been identified with genetic data (*Kunihiro & Reimer, 2018*).

The goal of the present study is to investigate the biodiversity of endosymbiotic microalgae found in meiofaunal marine acoels. Here, acoels containing microalgal symbionts were collected in subtropical localities in Southern Shikoku (Kochi) and in the Ryukyu Islands (Okinawa and Ishigaki), Japan. *Tetraselmis* and dinoflagellates (primarily in the family Symbiodiniaceae) were isolated from these acoels and were investigated using PCR (18S rRNA and 28S rRNA genes) and culture techniques. Results from this work aim to better understand acoel-microalgal symbiosis, by providing novel insights into acoel and microalgal biodiversity; we also aim to establish several new strains of microalgae that will improve our current and future understanding of microalgal systematics and taxonomy.

## MATERIALS AND METHODS

### Sample collection, light microscopy, and establishing microalgal cultures

Macroalgal and sediment samples were collected from intertidal and subtidal zones at seven sites in Okinawa, Ishigaki, and Kochi (Fig. 1 and Table S1) from April–October 2022. Both macroalgal and sediment samples were collected from all sites. Areas exposed to sunlight which had seagrass and corals in the vicinity were targeted. Samples were collected in plastic containers and transported back to the laboratory for further processing. Acoels were detached from macroalgae and sediment samples by replacing seawater with an isotonic magnesium chloride solution (*Schockaert, 1996*). The solution was stirred and then filtered through a 60–100 μm mesh. The mesh was then placed in a petri dish with filtered seawater and removed after 10–20 minutes. The contents of both the petri dish, and the material retained on the mesh, were then examined under an Olympus CKX53 (Olympus, Tokyo, Japan) inverted microscope and Olympus SZ61 (Olympus, Tokyo, Japan) stereomicroscope. Acoels were identified based on their planula body-shape, and other characteristics including the statocyst, eyespots, rhabdoid gland cells, and microalgal symbionts (*Achatz et al., 2013*; *Jondelius & Jondelius, 2020*). Using hand-drawn glass capillary tubes, acoels were washed multiple times with filtered seawater to eliminate any contaminants, and isolated into 24-well tissue culture plates containing filtered and autoclaved seawater. Acoels were imaged with either an Olympus CKX53

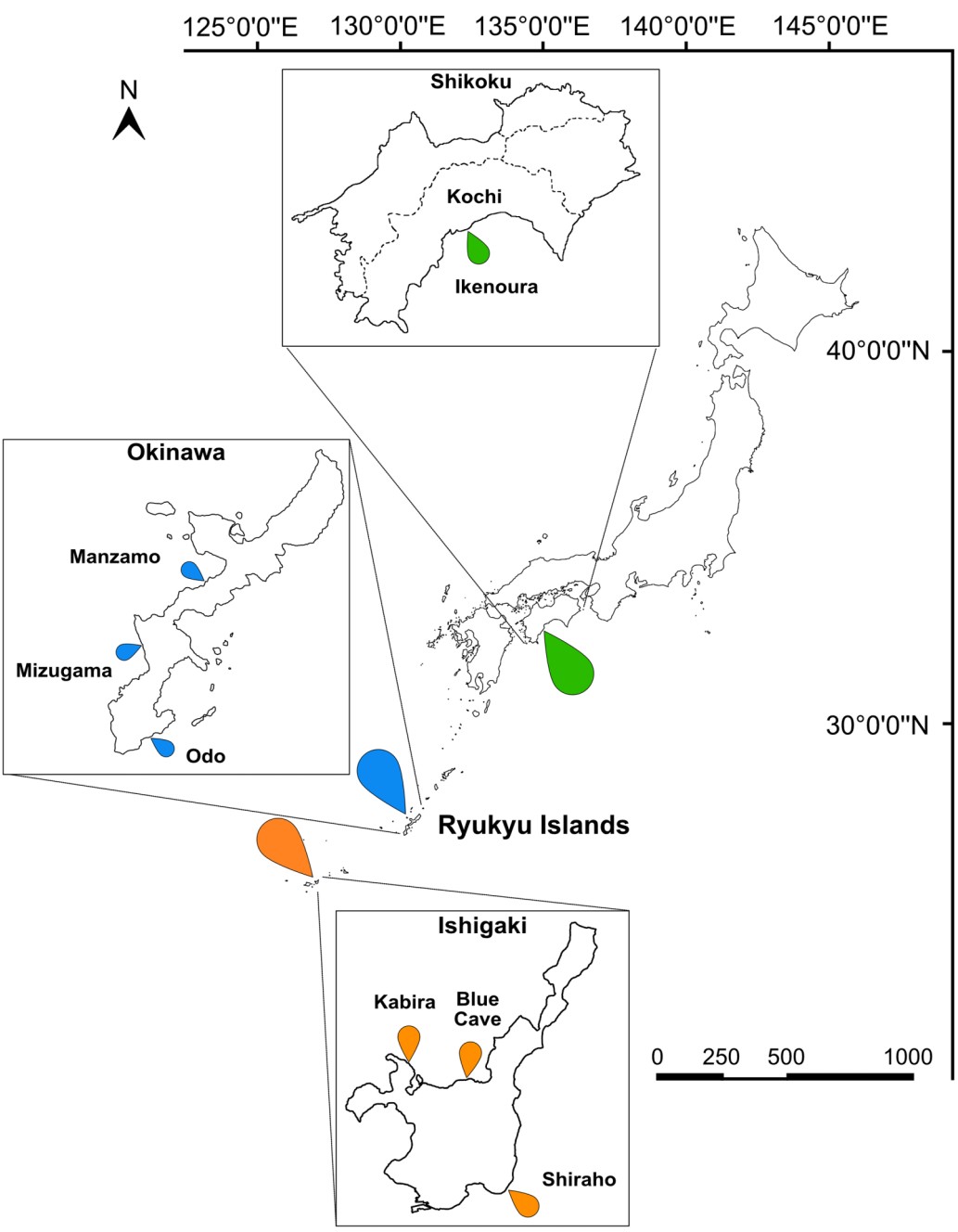

**Figure 1 Map of Japan indicating the sampling sites.** Sampling localities in Kochi (green marker), Okinawa (blue marker), and Ishigaki (orange marker) are indicated on the map and insets. Scale bar is shown in kilometers.

(Olympus, Tokyo, Japan), or Zeiss Axioskop 2 Plus (Zeiss, Oberkochen, Germany) connected to a Canon EOS Kiss X8i digital camera (Canon, Tokyo, Japan). Individual acoels were either preserved in 99.5% ethanol in 1.5 ml microcentrifuge tubes or broken apart under a dissection microscope and the internal microalgae were transferred to 24-well tissue culture plates. A small portion of acoels (isolates 22–29) were broken and washed in multiple wells to check for multiple symbionts. When cultures were visually

pigmented to the unaided eye, they were transferred to small petri dishes (*Tetraselmis*) or full-sized petri dishes (dinoflagellates). Microalgal cultures were maintained in half-concentration Daigo's IMK Medium (Wako Pure Chemical Industries, Tokyo, Japan) and were transferred to new dishes with fresh culture media every month. Where possible, after host acoels were broken apart, host material was transferred into a 1.5 ml microcentrifuge tube for DNA extraction. Cultures were maintained at 25 °C (12:12, light: dark hours) in Biotron Incubator LH-350 (NK Systems, Tokyo, Japan). Motile stages of *Tetraselmis* and Symbiodiniaceae were mounted on glass slides and imaged under a Zeiss Axioskop 2 Plus microscope (Zeiss, Oberkochen, Germany) connected to a Canon EOS Kiss X8i digital camera (Canon, Tokyo, Japan).

## DNA extraction, PCR amplification and sequencing

Genomic DNA from cultured microalgae and whole-acoel isolates was extracted according to manufacturer protocols with one of the following kits: DNeasy Blood & Tissue Kit (Qiagen, Hilden, Germany), QuickExtract FFPE DNA Extraction Kit (Lucigen, Middleton, WI, USA), or MasterPure Complete DNA and RNA Purification Kit (Epicentre, Madison, WI, USA) (Table S1).

For acoel isolates, nested PCR was performed on extracted DNA using general eukaryotic primers (Tables S1 and S2), and the products were cloned with an NEB PCR Cloning Kit (New England Biolabs, Tokyo, Japan) and then sequenced. For cultured microalgal material, nested PCRs were performed and then directly sequenced, or cloned and then sequenced. An additional *Tetraselmis* culture (K19), established from acoels collected in 2019 from Ikenoura, was also sequenced. All PCR reactions were first amplified using the SR1–28-1483R primer pair and then nested. From both the whole-acoel isolates and microalgal cultures, the 18S rRNA gene (henceforth, 18S) was amplified for *Tetraselmis* using SR1–18SRF and SR4–SR12 primer pairs, and a partial 28S rRNA gene (henceforth, 28S) was amplified for Symbiodiniaceae using the D1RF1–28-1483R primer pair. Apart from general eukaryotic primers, extracted host DNA was also amplified using a specific primer pair, Tear F–Tear R, designed from the sequences obtained from cloning. Polymerases used in the PCRs and thermocycler conditions are listed in Table S1; primers are listed in Table S2.

Following the nested reaction, PCR products were purified using Polyethylene Glycol (PEG) and directly sequenced or cloned. Acoel DNA inserts were screened by colony PCR using the S1512A–S1513A primer pair provided in the NEB cloning kit. Colony-PCR products were cut using a HAEIII restriction enzyme (Takara, Shiga, Japan) following the manufacturer's protocol. Digested products were checked on a 1% agarose gel. A minimum of four replicates of each banding pattern were selected. Select colony-PCR products were purified again with PEG, and then sequenced with BrilliantDye Terminator v3.1 (NimaGen, Nijmegen, Netherlands), according to manufacturer protocol on a 3130 genetic analyzer (Applied Biosystems, Waltham, MA, USA). Occasionally, additional sequencing primers were used to join sequence fragments (refer to Table S2).

### Phylogenic analyses of the 18S and 28S rRNA genes

Geneious Prime 2023.1.2 (https://www.geneious.com) was used to process Sanger sequencing reads. Cloning primers and low-quality regions (HQ% ≤ 50) were manually trimmed and assembled with the "*De Novo* Assemble" tool. *Tetraselmis* and dinoflagellate sequences were identified with NCBI's BLAST search tool, sorted into their respective data sets, and compiled with existing data from Genbank. Multiple Alignment using Fast Fourier Transform (MAFFT) (*Katoh et al., 2002*) within Geneious was then used to align both data sets. Alignments were manually trimmed, resulting in a final alignment of 1,807 bps and 1,391 bps for *Tetraselmis* 18S and Symbiodiniaceae 28S, respectively.

Iqtree v1.6.12 (*Nguyen et al., 2015*) was used to select the best-fit model for both Maximum-Likelihood and Bayesian analyses. Under the Akaike Information Criterion with correction (AICc), the models TN+F+R2 and TIM+3+F+R3 were selected for *Tetraselmis* and Symbiodiniaceae, respectively. Bootstrap analysis was performed on 1,000 pseudo-replicates. Under the Bayesian Information Criterion (BIC), the general time reversible (GTR) substitution model was selected. MrBayes 3.2.7a (*Huelsenbeck & Ronquist, 2001*) was used to calculate Bayesian Posterior Probabilities. Four Markov Chain Monte Carlo (MCMC) chains were run with the general time reversible (GTR) substitution model for 10,000,000 generations and sampled every 100th generation. The rate variation across sites were drawn from a gamma distribution with a proportion being invariable (lset rates = invgamma). Other parameters were set to the default settings. Chain convergence was checked with Tracer v1.7 and the posterior probability values were obtained from the consensus tree after the first 25% was discarded as burn-in. Molecular Evolutionary Genetics Analysis version 11 (MEGA11) (*Tamura, Stecher & Kumar, 2021*) was used to compute the uncorrected pairwise distance for Symbiodiniaceae sequences. Default settings were used: no variance estimation method, uniform rates among sites, and complete deletion of gaps and missing data.

Sequences generated from this study were deposited to NCBI. Accession numbers are listed in Table S1.

## RESULTS

### Sampling and identification of host acoels

A total of 66 acoels were sampled from seven sites (Fig. 1; Table S1). Acoels consisted of four main morphotypes that were differentiated based on symbionts, as well as features including ocelli, rhabdoid type, and body shape. The first (Fig. 2A) and second (Fig. 2B) morphotype harbored exclusively *Tetraselmis*. These were distinguishable, based on the numerous blunt-ended colored rhabdoids seen only in the second morphotype. A third type of acoel harbored exclusively Symbiodiniaceae and did not have any visible ocelli or colored rhabdoids (Fig. 2C). The fourth type were acoels collected from Ikenoura that contained *Tetraselmis* and *Amphidinium*. These also possessed three posterior caudal lobes (Fig. 3).

Of the 66 acoels collected, 36 contained exclusively *Tetraselmis* symbionts (Figs. 4–6); 26 contained exclusively Symbiodiniaceae (dinoflagellate) symbionts (Figs. 7 and 8); one

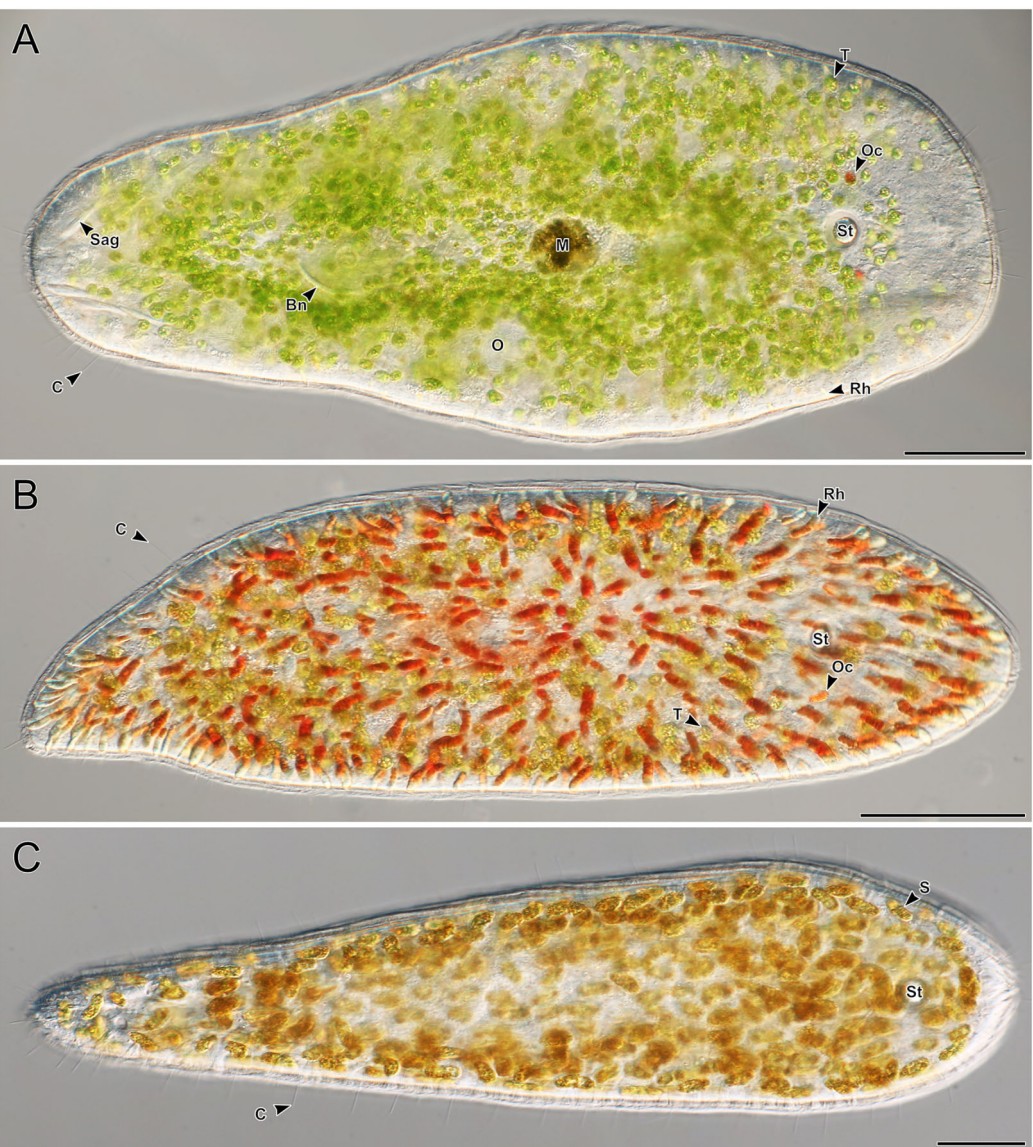

**Figure 2 Representative light micrographs of live meiofaunal marine acoels.** (A–C) Dorsal view of live acoels. Individuals are oriented with the anterior end pointed to the right. (A) Acoel harboring *Tetraselmis* symbionts (T) with ocelli (Oc) on both sides of the statocyst (St), pigmentation near the mouth (M), a paired bursal nozzle (Bn), developing oocytes (O), cilia (C), and two types of rhabdoid gland cells: sagittocysts (Sag), and translucent pigmented rhabdoid cells (Rh). (B) Acoel harboring *Tetraselmis* symbionts (T) with ocelli (Oc) on both sides of the statocyst (St), cilia (C), and numerous red orange rhabdoid gland cells (Rh) distributed throughout the parenchyma. (C) Acoel harboring Symbiodiniaceae symbionts (S), with a statocyst (St), cilia (C), but no ocelli or colored rhabdoid gland cells. Scale bars: 100 μm.

contained both Symbiodiniaceae and *Amphidinium* symbionts (Fig. 8B); and three contained a mix of *Tetraselmis* and *Amphidinium* symbionts (Fig. 3).

Maximum-Likelihood analysis of 23 host acoel 18S sequences (Fig. 9) formed three genetic groups that corresponded with three different morphotypes: the first and second morphotype within Convolutidae and the third morphotype comprising a clade with

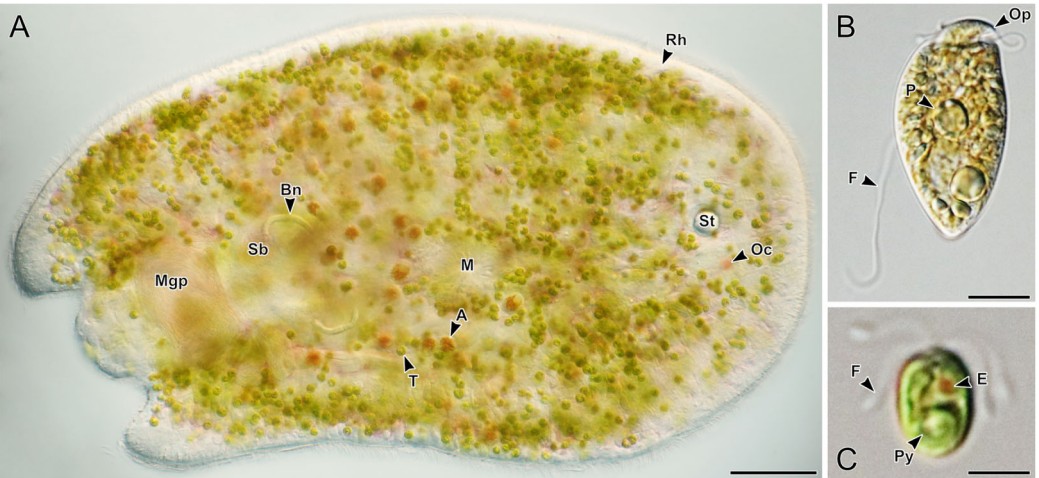

**Figure 3 Representative light micrographs of live acoels collected from Ikenoura, Kochi and their cultured *Tetraselmis* and *Amphidinium* endosymbionts.** (A) Dorsal view of an individual containing *Tetraselmis* (T) and *Amphidinium* (A) endosymbionts. Internally, several features including the male gonopore (Mgp), seminal bursal (Sb), bursal nozzle (Bn), mouth (M), rhabdoid gland cells (Rh), statocyst (St), and ocelli (Oc) were visible. (B) General morphology of *Amphidinium gibbosum* in an established culture (K22). The operculum (Op), pyrenoid (P), and flagella (F) are shown. (C) General morphology of *Tetraselmis* closely related to *T. convolutae* in an established culture. A single pyrenoid (Py), red eyespot (E), and two pairs of flagella (F) are shown. Scale bar: 100 μm (A); 5 μm (B and C).

Mecynostomidae. Genetic data were not obtained from 43 out of 62 acoel hosts, likely due to host DNA being lost while breaking apart host tissue to free algal symbionts.

No sequences for the fourth morphotype were obtained. Isolates within Convolutidae with corresponding genetic data harbored either *Tetraselmis* or Symbiodiniaceae symbionts, while isolates sister to Mecynostomidae harbored exclusively Symbiodiniaceae symbionts. Information on the isolate number, symbiont type, locality, and dates can be found in Table S1.

## Cultivation of *Tetraselmis* and dinoflagellate microalgal strains

In the majority of cases, a single microalgal culture was established from one host acoel. Exceptions to these included isolates 19–21 collected from Ikenoura, isolate 61 (Fig. 8B) from Shiraho, and isolates 27 and 28 (Figs. 7H and 7J) from Kabira. Isolates 19–21 contained two types of symbionts, *Tetraselmis* sp. and *Amphidinium gibbosum* (Fig. 3). Cultures were established for both microalgae. Isolate 61 contained *Symbiodinium* sp. and *A. gibbosum*; but only *Symbiodinium* sp. was successfully cultured. Isolate 27 contained *Cladocopium* and *Symbiodinium* ($A_{IV}$), and isolate 28 contained *Symbiodinium*, subclades $A_{III}$ and $A_{IV}$; all symbionts were cultured. In total, 32 *Tetraselmis* cultures, 26 Symbiodiniaceae cultures, and two *Amphidinium* cultures were established (and sequenced) from acoels (Fig. S2).

## Molecular diversity of *Tetraselmis*

Maximum-likelihood analysis of *Tetraselmis* 18S sequences (Fig. 10) formed four main groups that clustered around *T. convolutae*, *T. astigmatica*, *T. marina*, and *T. gracilis*.

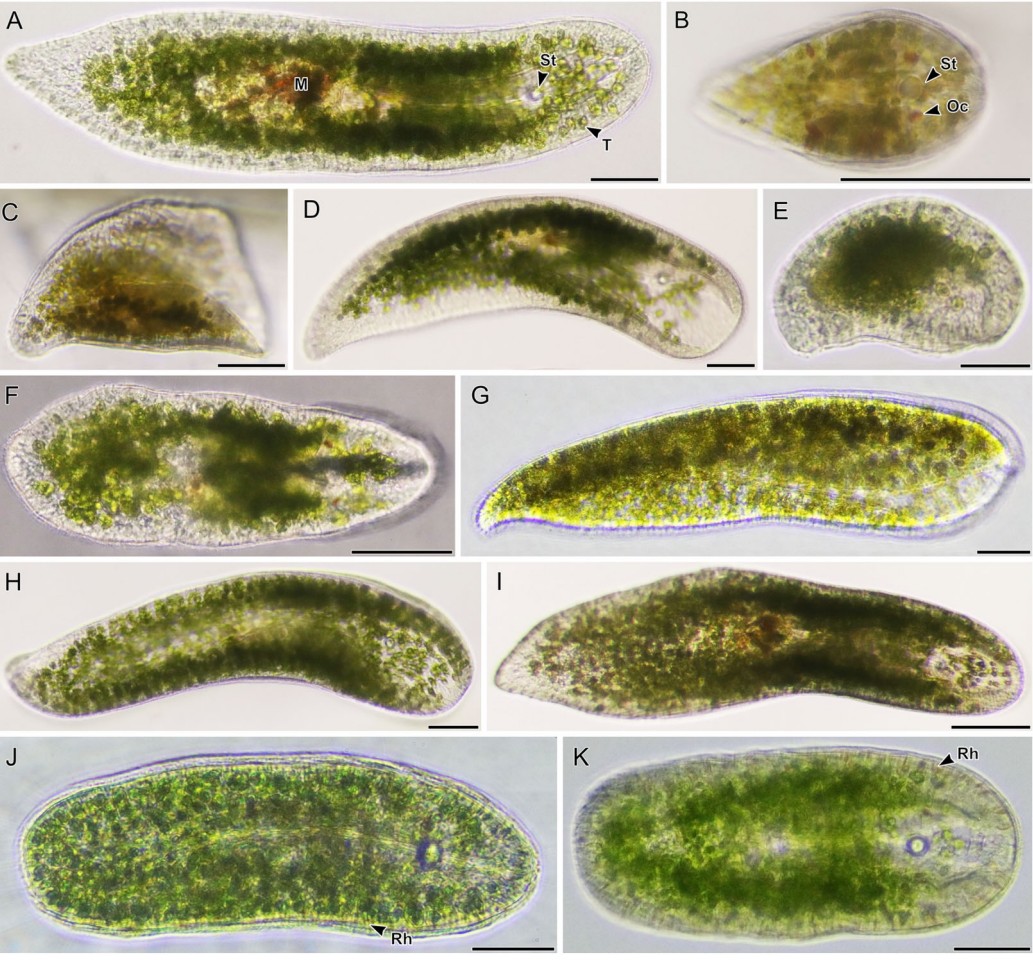

**Figure 4 Light micrographs of live acoels containing *Tetraselmis* symbionts.** (A–I) Acoels harboring symbiotic microalgae from the *Tetraselmis astigmatica* clade. Acoels are oriented with the anterior to the right. (A) Isolate 4 collected from Kabira showing *Tetraselmis* symbionts (T), pigmentation surrounding the mouth (M), and the statocyst (St). (B) Isolate 1 collected from Shiraho, a juvenile acoel with ocelli (Oc) located on both sides of the statocyst (St). (C) Isolate 17 collected from Mizugama. (D) Isolate 44 collected from Kabira. (E) Isolate 9 collected from Kabira. (F) Isolate 8 collected from Kabira. (I) Isolate 3 collected from Kabira. (J and K) Isolates 58 and 59 collected from Shiraho harboring *Tetraselmis* closely related to *T. convolutae* with rhabdoid gland cells (Rh). Scale bars: 100 μm.

The most often encountered *Tetraselmis* was clustered with *T. astigmatica* (26 cultures, and eight sequences from total-host DNA), followed by *T. convolutae* (four cultures, and three sequences from total-host DNA), *T. marina* (two cultures), and *T. gracilis* (one culture). Deeper nodes in the *Tetraselmis* phylogenetic analyses were largely unresolved.

## Molecular diversity of Symbiodiniaceae

Maximum-Likelihood analysis of Symbiodiniaceae 28S sequences (Fig. 11) formed six clusters across three different clades: *Miliolidium* (one sequence from total-host DNA), *Cladocopium* (three cultures), and *Symbiodinium* (23 cultures, and eight sequences from total-host DNA). The *Miliolidium* clade included isolate 16 (Fig. 7E) that grouped with five

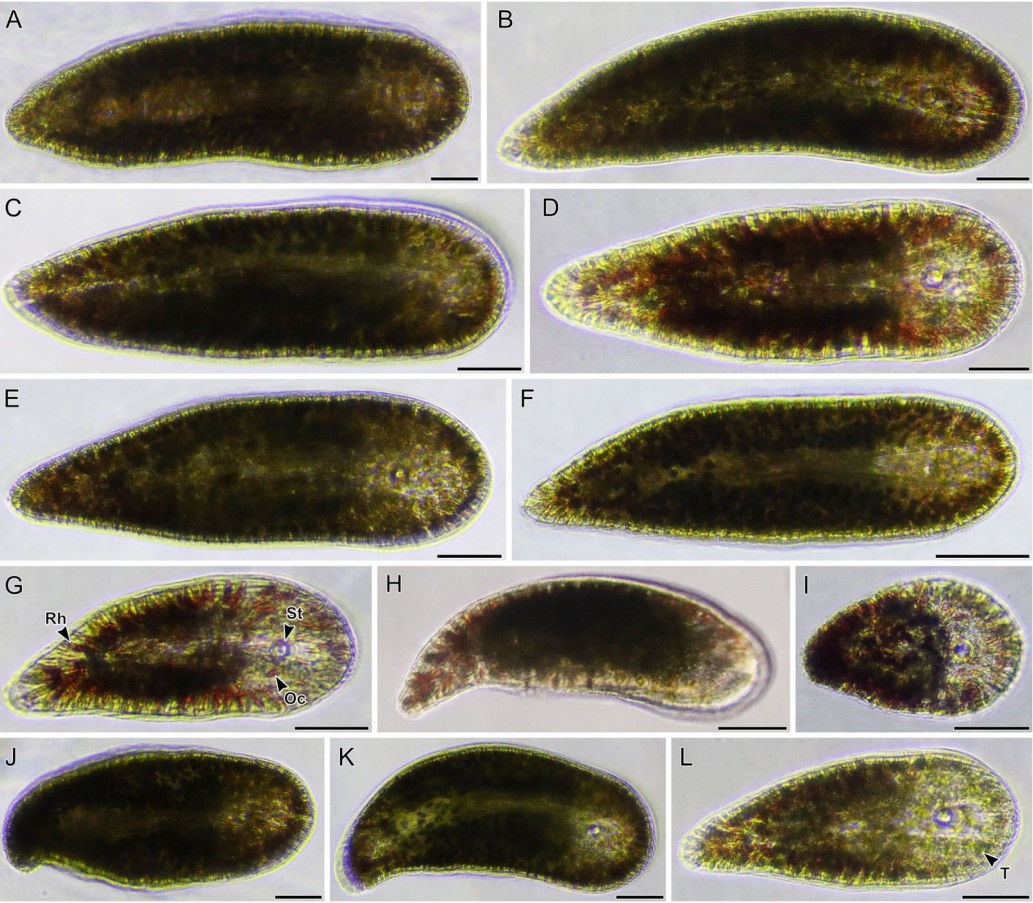

**Figure 5 Light micrographs of live acoels with red rhabdoid gland cells in the parenchyma containing *Tetraselmis* symbionts.** (A–L) Acoels harboring symbiotic microalgae from the *Tetraselmis astigmatica* clade. Acoels are oriented with the anterior to the right. (A) Isolate 36 collected from Kabira. (B) Isolate 35 collected from Kabira. (C) Isolate 50 collected from Kabira. (D) Isolate 49 collected from Kabira. (E) Isolate 54 collected from Kabira. (F) Isolate 34 collected from Kabira. (G) Dorsal view of isolate 48 collected from Kabira, a juvenile acoel with red rhabdoid cells (Rh), a statocyst (St), and ocelli (Oc). (H) Isolate 18 collected from Odo. (I) Isolate 52 collected from Kabira. (J) Isolate 45 collected from Kabira. (K) Isolate 46 collected from Kabira. (L) Isolate 37 collected from Kabira, a juvenile acoel with visible *Tetraselmis astigmatica* symbionts (T). Scale bars: 100 μm.

other sequences isolated from Foraminifera and Porifera. The (uncorrected pairwise) distance between the type species, *Miliolidium leei Pochon & LaJeunesse, 2021*, and the *Miliolidium* sequence obtained in the present study was 0.5%. The *Cladocopium* clade included strains 27.1–27.3 from isolate 27 which were sister to *C. infistulum*, *C. thermophilum*, and *C. goreaui*. The intra-clade distances between our sequences and these species ranged from 1–4%. These strains had a 1.6% difference to the type species, *Cladocopium goreaui*. Many sequences in *Symbiodinium* grouped together with *S. natans* (13 cultures, and two from total-host DNA); the genetic distance did not exceed 1%. The remaining sequences formed a sister group with either *S. tridacnidorum* (nine cultures), *S. microdriaticum* (one culture and four sequences from total-host DNA extractions), or *S. pilosum* (two sequences from total host DNA extractions); genetic

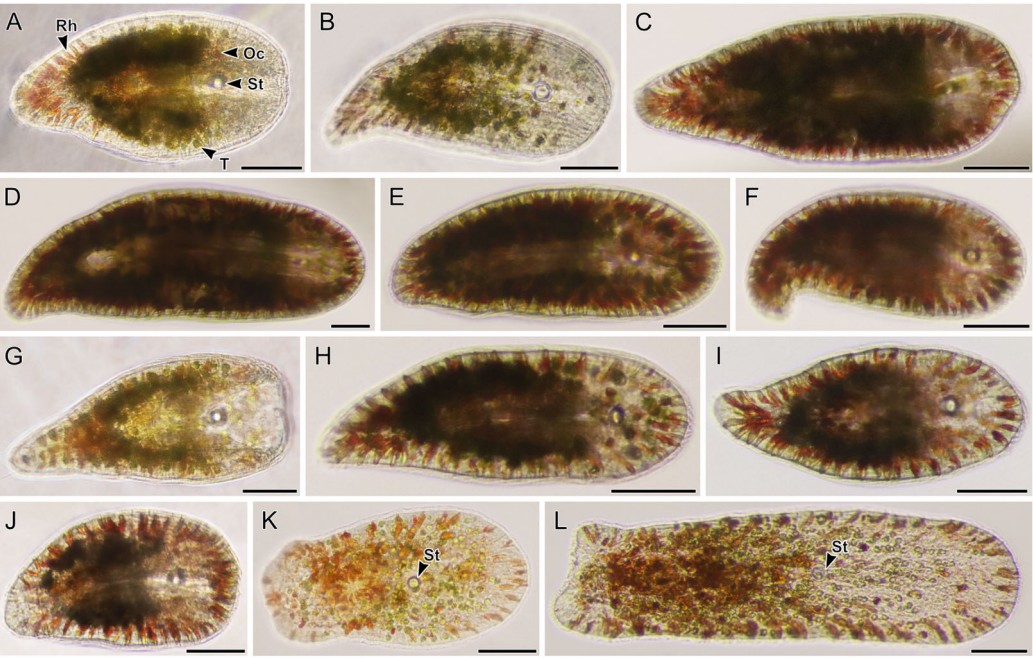

**Figure 6 Light micrographs of live acoels with red rhabdoid gland cells in the parenchyma containing *Tetraselmis* symbionts.** (A–J) Acoels harboring symbiotic microalgae from the *Tetraselmis astigmatica* clade. Acoels are oriented with the anterior to the right. (A) Isolate 22 collected from Odo, a juvenile acoel showing the red rhabdoid gland cells (Rh), statocyst (St), ocelli (Oc), and *Tetraselmis astigmatica* symbionts (T). (B) Isolate 25 collected from Odo. (C) Isolate 53 collected from Kabira. (D) Isolate 41 collected from Kabira. (E) Isolate 40 collected from Kabira. (F) Isolate 39 collected from Kabira. (G) Isolate 24 collected from Odo. (H) Isolate 32 collected from Kabira. (I) Isolate 31 collected from Kabira. (J) Isolate 23 collected from Odo. (K and L) Isolates 57 and 5 collected from Kabira, possessing a blunt posterior end and a statocyst (St) towards the middle of the body and containing algal symbionts from the *Tetraselmis marina* clade. Scale bars: 100 μm.

distances between our sequences and sequences they grouped with ranged between 0.5–2%, 2–3%, and 4–6%, respectively.

### Phylogeny of other dinoflagellates sequenced from acoels

In addition to Symbiodiniaceae, the dinoflagellate *Amphidinium* was also observed (isolates 19–21) and cultured (isolates 61 and K22) from acoels. *Amphidinium* cultures were identified to the genus level based on the presence of an operculum. The 28S sequences from these *Amphidinium* isolates, 61 and K22, clustered with *A. gibbosum* (Fig. S1), including an *A. gibbosum* from the acoel *Waminoa* sp. (AB626894). Additionally, a *Heterocapsa* sequence (isolate 7) was sequenced, and was positioned sister to *H. pseudotriquetra* (Fig. S3). It should be noted, however, that *Heterocapsa* was not cultured or directly observed within acoels during this study.

## DISCUSSION

### Meiofaunal marine acoels harbor diverse lineages of *Tetraselmis*

In this study, we found at least four distinct lineages of *Tetraselmis* associated with marine acoels. Several *Tetraselmis* from Kochi and Ishigaki (isolates 19–21, and cultures K19, K22,

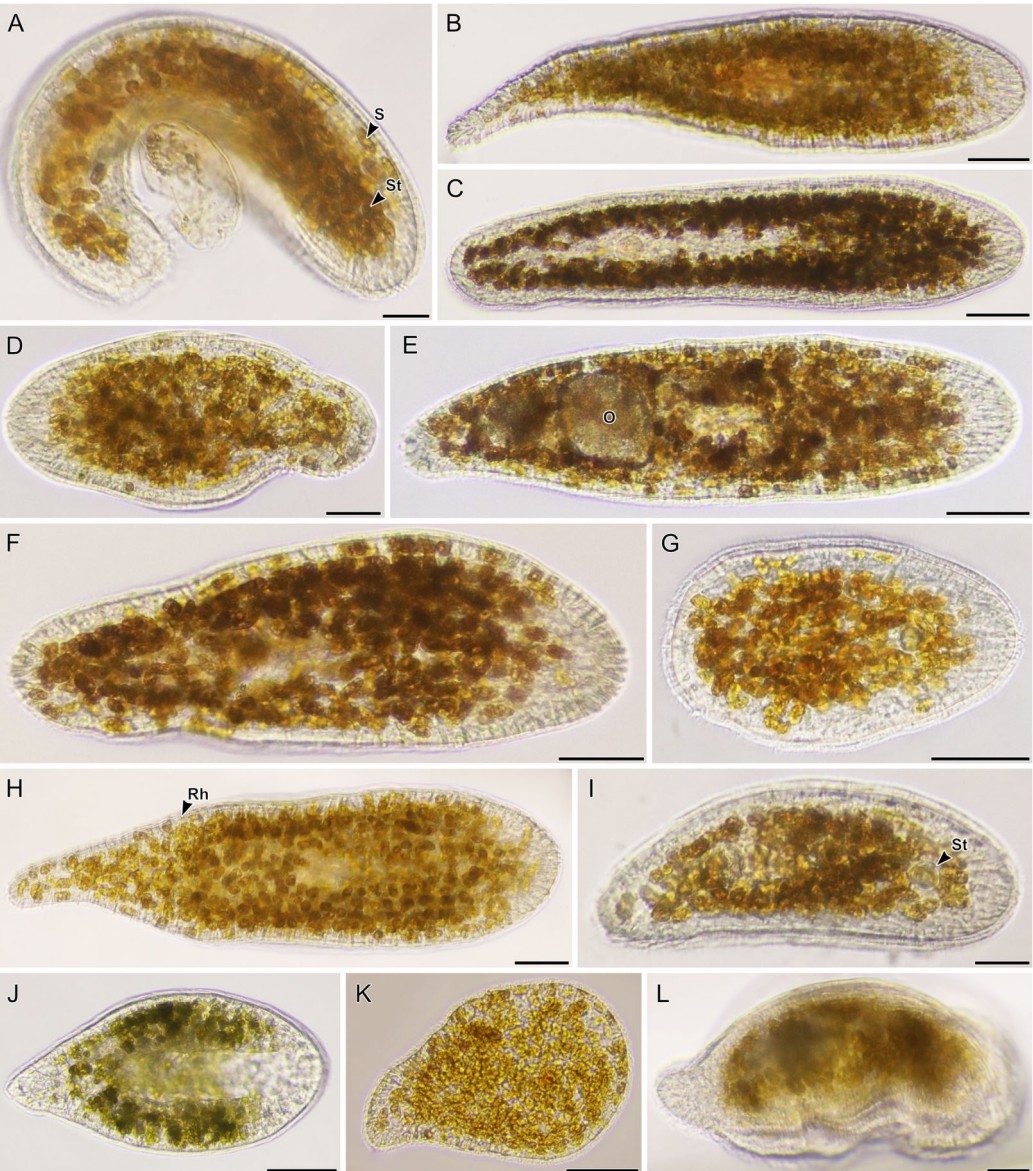

**Figure 7 Light micrographs of live acoels containing dinoflagellates from the family Symbiodiniaceae.** (A–L) Acoels harboring dinoflagellates from the family Symbiodiniaceae. Acoels are oriented with the anterior to the right. No individuals, juvenile or adult, were observed with ocelli or colored rhabdoid gland cells. (A) Isolate 10 collected from Blue Cave containing a statocyst (St) at the anterior end and *Symbiodinium* cf. *natans* endosymbionts (S). (B) Isolate 13 collected from Manzamo. (C) Isolate 2 collected from Kabira. (D) Isolate 12 collected from Manzamo. (E) Isolate 16 collected from Manzamo containing an oocyte (O), and harboring *Miliolidium* sp. endosymbionts. (F) Isloate 26 collected from Kabira. (G) Isloate 14 collected from Manzamo. (H) Isolate 27 collected from Kabira containing both *Cladocopium* and *Symbiodinium* endosymbionts. Rhabdoid gland cells (Rh) are needle shaped and translucent, observed to be evenly distributed along the parenchyma. (I) Isolate 15 collected from Manzamo, a juvenile with statocyst (St) visible. (J) Isolate 28 collected from Kabira. (K) Isolate 29 collected from Kabira. (L) Isolate 11 collected from Manzamo. Scale bars: 100 μm.

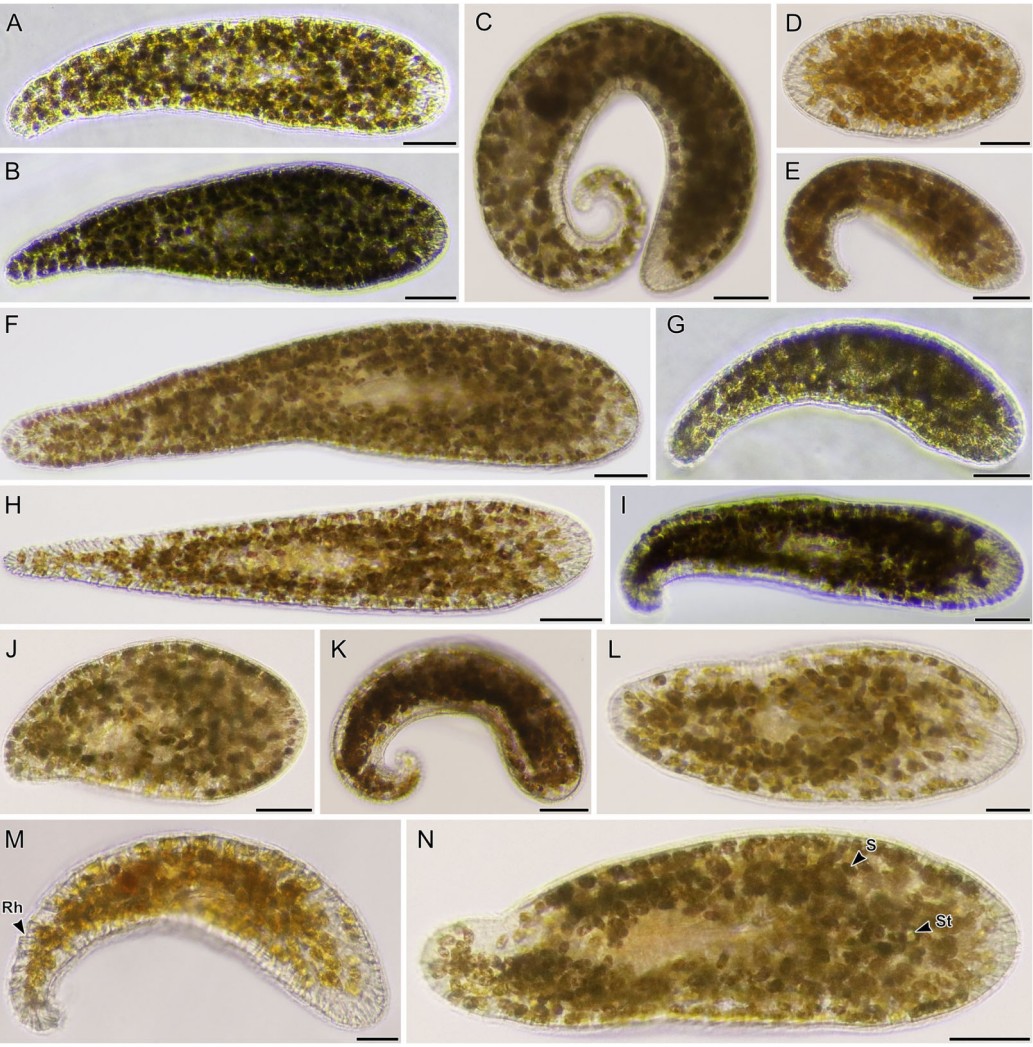

**Figure 8 Light micrographs of live acoels containing dinoflagellates from the family Symbiodiniaceae.** (A–N) Acoel harboring dinoflagellates from the family Symbiodiniaceae. Acoels are orientated with the anterior to the right. (A) Isolate 63 collected from Shiraho. (B) Isolate 61 collected from Shiraho containing *Symbiodinium* and *Amphidinium*. (C) Isolate 42 collected from Kabira. (D) Isolate 62 collected from Shiraho. (E) Isolate 38 collected from Kabira. (F) Isolate 43 collected from Kabira. (G) Isolate 65 collected from Shiraho. (H) Isolate 30 collected from Kabira. (I) Isolate 64 collected from Shiraho. (J) Isolate 33 collected from Kabira. (K) Isolate 60 collected from Shiraho. (L) Isolate 56 collected from Kabira. (M) Isolate 66 collected from Shiraho showing translucent rhabdoid gland cells (Rh) distributed along the parenchyma. (N) Isolate 47 collected from Kabira showing a statocyst (St) and its Symbiodiniaceae symbionts (S). Scale bars: 100 μm.

58, and 59) were closely related to *T. convolutae* from *Symsagittifera roscoffensis* (*Parke & Manton, 1967*). *Tetraselmis convolutae* strains from *S. roscoffensis* from Roscoff, France (RCC1563 and RCC1564) are the only *Tetraselmis* to be formally described from acoels (*Parke & Manton, 1967*). The novel Japan isolates from this study formed a sister clade to *T. convolutae*, but statistical support at shared nodes was weak. Whether *Tetraselmis* strains from this study are the same species as *T. convolutae* (from Roscoff) is difficult to

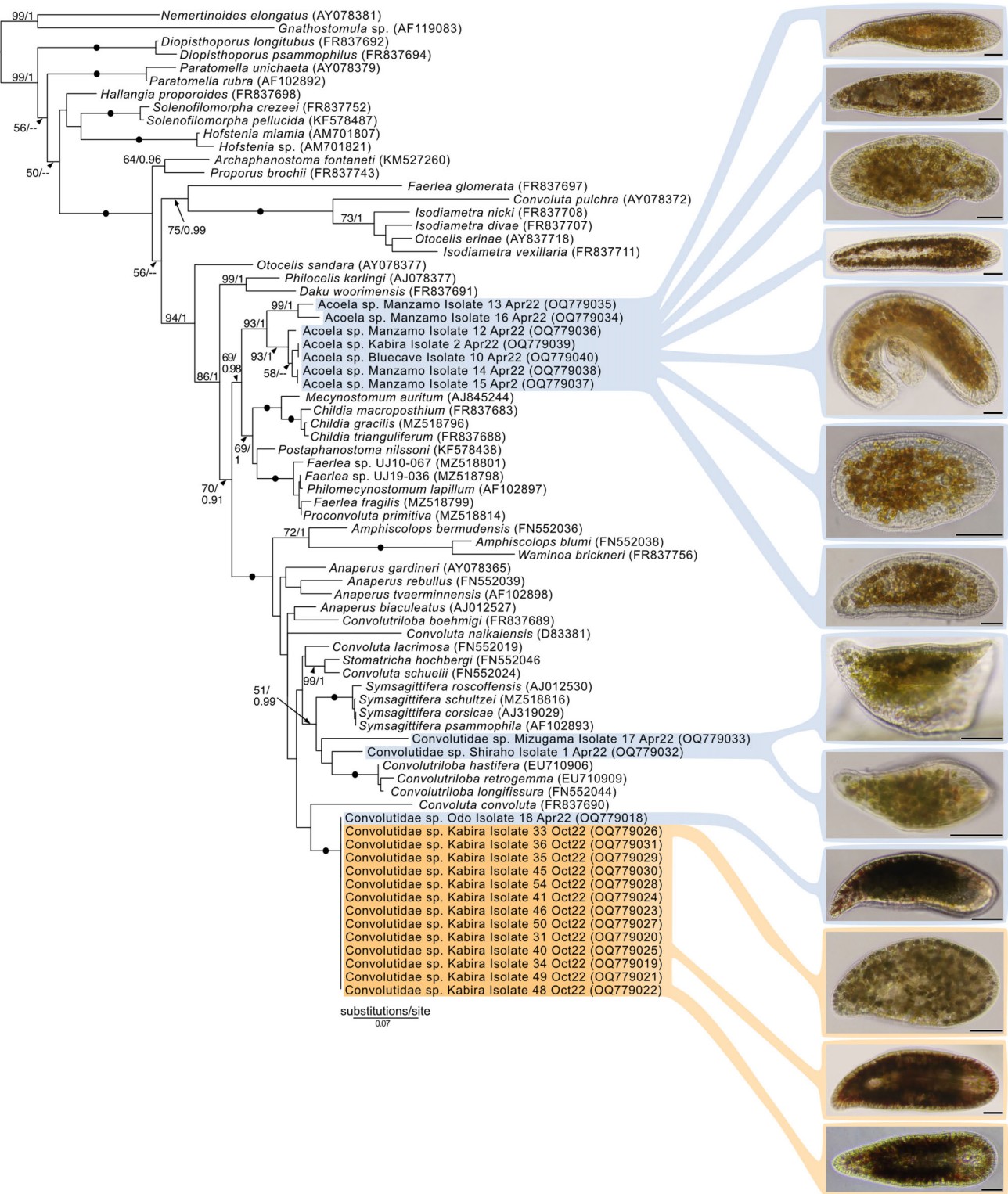

**Figure 9  Maximum-Likelihood (ML) tree of Acoelomorpha inferred from the 18S rRNA gene.** Sequences generated in this study are highlighted in colors that denote sampling times: April (blue), May (green), July (pink), and October (orange). Maximum-Likelihood bootstrap values <50 and Bayesian Posterior Probabilities (BPP) <0.90 were omitted. Black dots indicate fully supported branches (100 ML/1.00 BPP). Long branches are shortened by multiples of the substitutions/site scale bar (indicated on the branch). Representative light micrographs of acoels are shown on the right. Scale bars: 100 μm.                                           

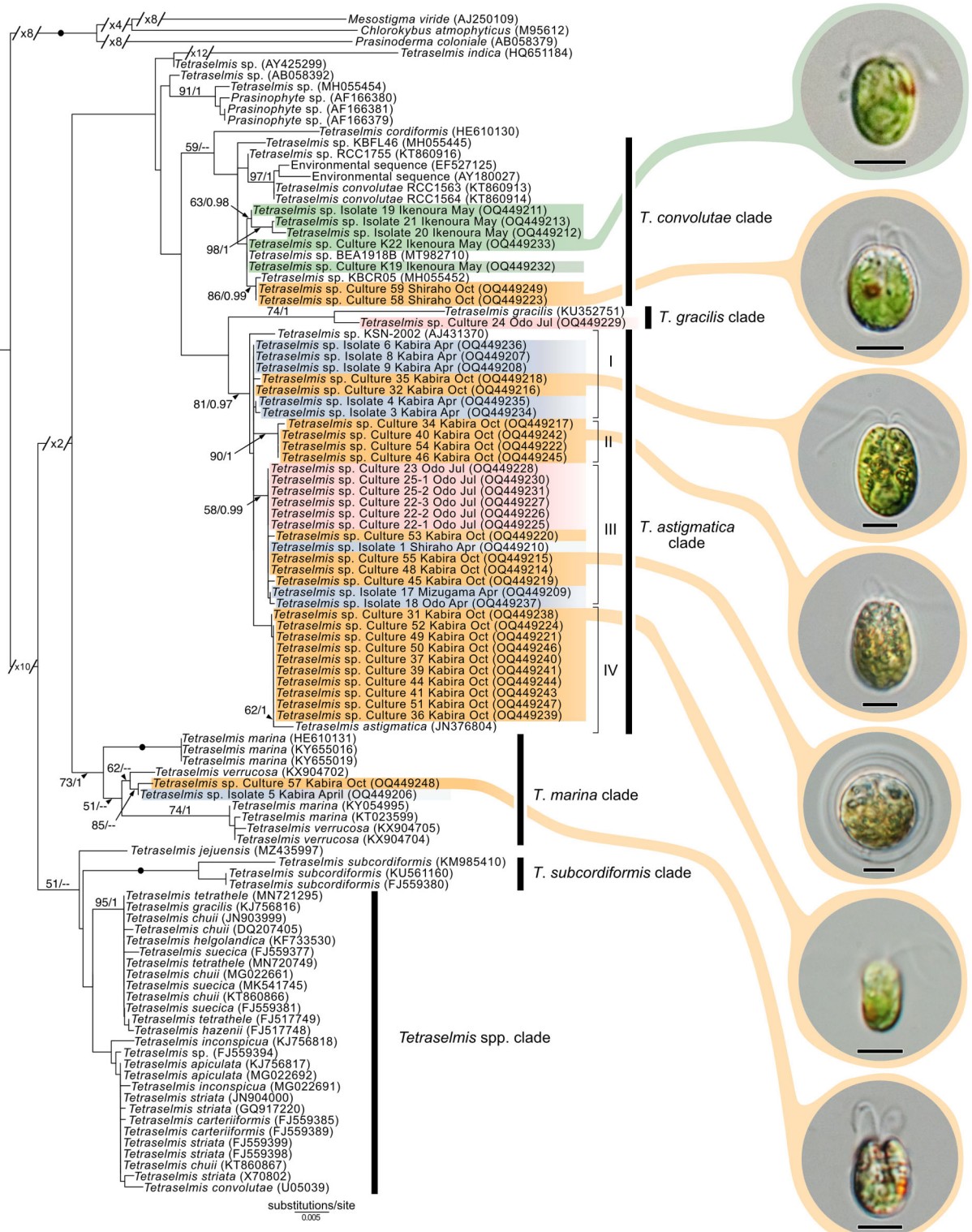

**Figure 10  Maximum-Likelihood (ML) tree of *Tetraselmis* inferred from the 18S rRNA gene.** Sequences generated in this study are highlighted in colors that denote sampling times: April (blue), May (green), July (pink), and October (orange). Maximum-Likelihood bootstrap values <50 and Bayesian Posterior Probabilities (BPP) <0.95 were omitted. Black dots indicate fully supported branches (100 ML/1.00 BPP). Long branches are shortened by multiples of the substitutions/site scale bar (indicated on the branch). Representative light micrographs of microalgae from different cultures are connected to their respective lineages. Scale bars: 5 μm.                

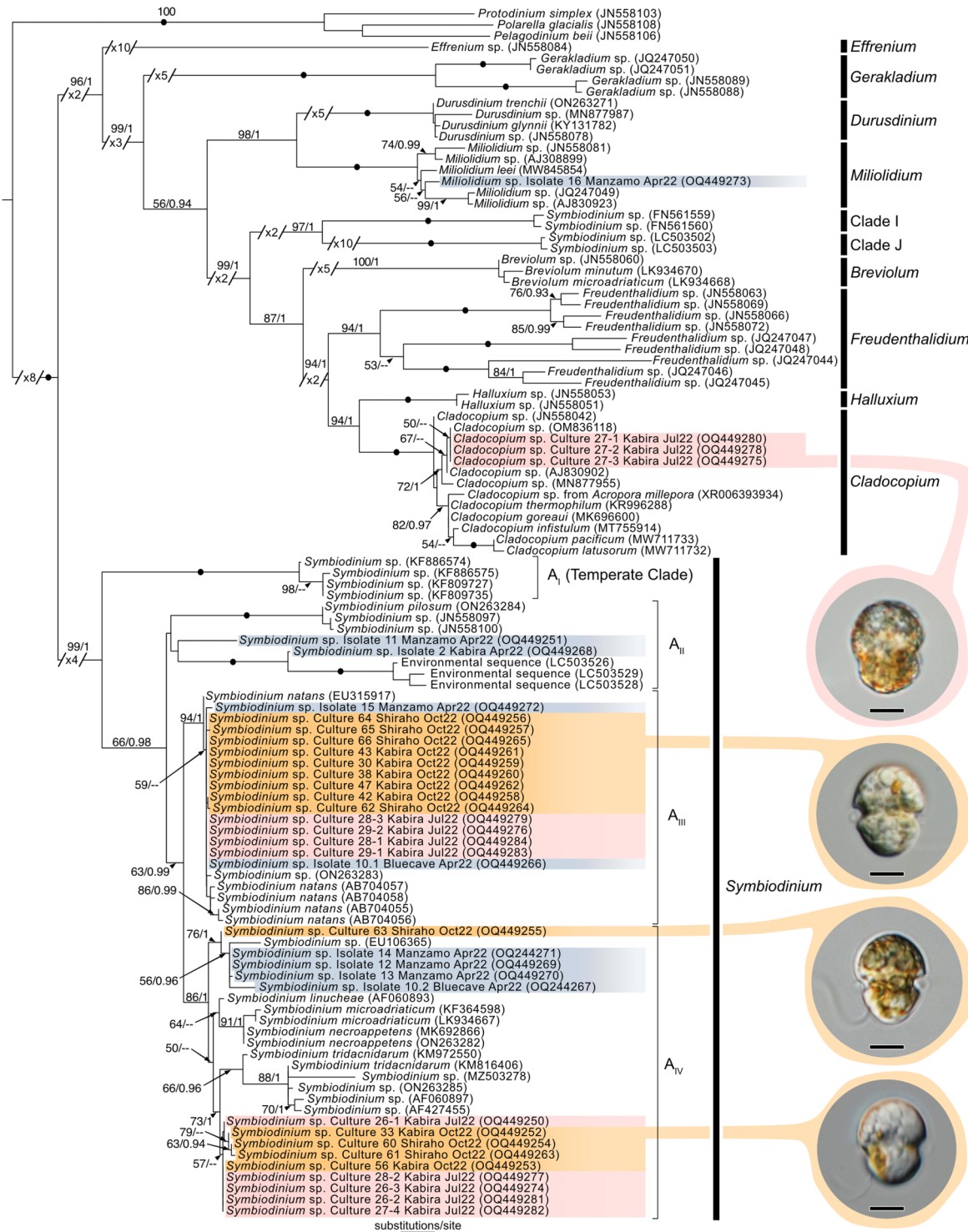

**Figure 11 Maximum-Likelihood (ML) tree of Symbiodiniaceae inferred from the 28S rRNA gene.** Sequences generated in this study are highlighted in colors that denote sampling times: April (blue), May (green), July (pink), and October (orange). Maximum-Likelihood bootstrap values <50 and Bayesian Posterior Probabilities (BPP) <0.95 were omitted. Black dots indicate fully supported branches (100 ML/1.00 BPP). Long branches are shortened by multiples of the substitutions/site scale bar (indicated on the branch). Representative light micrographs of microalgae from different cultures are connected to their respective lineages. Note: Clade I and Clade J are based on results from *Pochon & Gates (2010)*, and *Yorifuji et al. (2021)*, respectively. Scale bars: 5 μm.

conclude based on 18S data alone. Additional morphological data are needed to make a distinction between these closely related strains.

Apart from *T. convolutae*, we report members of the *T. marina*, *T. gracilis*, and *T. astigmatica* clades as symbionts of acoels for the first time. Specifically, isolate 5 and culture 57 from Kabira were closely related to *T. verrucosa* (KX904702), a free-living strain collected from Messolonghi, Greece (*Chantzistrountsiou et al., 2016*). Both Japan isolates were well-nested within the *T. marina* clade (Fig. 10), with a genetic distance ranging from 0.3–1.1%. The red-orange coloration indicated that culture 57 is *T. verrucosa* f. *rubens*, however, measurements of both the position and distribution of the chloroplast is needed to identify these isolates (*Chantzistrountsiou et al., 2016*; *Hori, Norris & Chihara, 1983*). Free-living *T. marina* and *T. verrucosa* f. *rubens* have been described (with morphological data) from Japan (*Hori, Norris & Chihara, 1983*). The present study provides the first molecular data of these Japanese lineages. Isolate 24 from Odo was positioned sister to *T. gracilis* (KU352751), a free-living strain collected from the Arabian Sea (unpublished NCBI data). However, another unpublished NCBI sequence, *T. gracilis* (KJ756816), was positioned in a separate clade and represents a strain from the North Sea (*Butcher, 1959*). Thus, the identity of isolate 24 from Odo, Japan remains uncertain though it is likely distinct from the *T. gracilis* described by *Butcher (1959)*.

The majority of *Tetraselmis* sequences generated in this study clustered within the *T. astigmatica* clade. Isolates from multiple sites around Okinawa and Ishigaki tended to form multiple subclades (Fig. 10, I—IV). Currently, only two sequences that represent *T. astigmatica* exist (KSN-2002 and CCMP880). The strain CCMP880 (JN376804) is presumably the same strain used in the description of *T. astigmatica* (*Hori, Norris & Chihara, 1982*, *1983*, *1986*; *Norris, Hori & Chihara, 1980*). Both sequences are from unpublished sources. The topology of the tree shows multiple subclades closely related to *T. astigmatica*, but molecular phylogenetic analysis based on 18S alone lacks the resolution to clearly resolve these (*Piganeau et al., 2011*). Of note are inconsistencies found between light micrographs taken of *Tetraselmis* strains (culture 31, Fig. S2D) and the original *T. astigmatica* description (*Hori, Norris & Chihara, 1982*). The most intriguing of these is the presence of a stigma, a feature absent in the description of *T. astigmatica* by *Hori, Norris & Chihara (1982)*. A more detailed investigation utilizing additional genetic markers (ITS, *psb*A. and *rbc*L), as well as ultrastructural data, is warranted to resolve what might be a species complex. Nonetheless, this study is the first to report lineages in *T. marina*, *T. gracilis*, and *T. astigmatica* as symbionts of acoels.

## Reconciling novel molecular data into the contemporary taxonomic framework of *Tetraselmis*

Assigning species names to molecular clades and reconciling novel (molecular) data within this genus remains difficult. Classification of *Tetraselmis* species is based on the morphological characteristics of cells including symmetry, presence/position of the stigma, and the ultrastructure of the pyrenoid and flagella (*Butcher, 1959*; *Hori, Norris & Chihara, 1982*, *1983*, *1986*; *Marin, Matzke & Melkonian, 1993*; *Norris, Hori & Chihara, 1980*). Currently, molecular data (18S) is not available for over half the accepted (~30) *Tetraselmis*

species. Only three species (*T. jejuensis*, *T. indica*, and *T. verrucosa*) have type strains for which both molecular and morphological data is available (*Arora et al., 2013*; *Chantzistrountsiou et al., 2016*; *Hyung et al., 2021*). Moreover, different sequences with the same name (*T. convolutae*, *T. gracilis*, and *T. chuii*) appear in different clades, further clouding *Tetraselmis* taxonomy (*Arora et al., 2013*; *Hyung et al., 2021*). This study established 32 cultures of *Tetraselmis* from acoels providing a vital resource to address fundamental issues surrounding *Tetraselmis* taxonomy and systematics.

## Marine acoels also contain a diversity of dinoflagellate microalgae

The majority of dinoflagellate sequences generated from acoels in this study belonged to the family Symbiodiniaceae. Identification of these symbionts in acoels (except for those in *Waminoa*) has been completely based on observational data and lacks corresponding molecular information (*Achatz, 2008*; *Achatz & Hooge, 2006*; *Hooge & Smith, 2004*; *Yamasu, 1982*). *Kunihiro & Reimer (2018)* found that *Waminoa* in Okinawa contained only a single lineage of *Cladocopium* but speculated that acoels harbor a much more diverse range of Symbiodiniaceae.

Novel (28S) sequences generated in this study formed six clusters across *Symbiodinium*, *Cladocopium*, and *Miliolidium*: four in *Symbiodinium*, and one each in *Cladocopium* and *Miliolidium* (Fig. 11). This study is the first report of four lineages of *Symbiodinium* and *Miliolidium*, as symbionts of acoels. Within *Symbiodinium*, one cluster was positioned in subclade AIII; the sequences were grouped with *S. natans*. The presence of *S. natans* in cultures established from cnidarian or other metazoan hosts has been attributed to their abundance in the environment (*e.g.,* contamination) or possible background symbiosis. They have also not been established as a symbiont under experimental conditions (*LaJeunesse et al., 2018*). This is intriguing, as acoels may be a possible reservoir, serving as a primary host for *S. natans*.

The remaining three clusters, one in subclade AII and two in subclade AIV, could represent novel lineages. These latter three formed separate branches with moderate support. The presence of *S. natans* in cultures established from cnidarian or other metazoan hosts has been attributed to their abundance in the environment (*e.g.,* contamination) or possible background symbiosis. They have also not been established as a symbiont under experimental conditions (*LaJeunesse et al., 2018*). This is intriguing, as acoels may be a possible reservoir, serving as a primary host for *S. natans*. Other *Symbiodinium* sequences from this study did not group with known species in the phylogenetic analyses. The identity (or novelty) of these *Symbiodinium* strains will have to be confirmed through sequencing of a more variable genetic region (*e.g.,* ITS), and by morphological comparisons.

*Cladocopium* sequences (isolate 27) grouped with other undescribed species, separate from *C. goreaui* and other described coral symbionts within that same clade. Our *Cladocopium* sequences (cultures 27.1–27.3) could not be compared with symbionts of *Waminoa* acoels—because different genes (ITS-2 and psbAncr) were utilized (*Kunihiro & Reimer, 2018*). It is important to note that a distinct strain of *Symbiodinium* (culture 27.4) was also established, along with *Cladocopium*, from isolate 27. This potentially shows some

adaptability in acoels, like corals that also host a variety of Symbiodiniaceae at the same time.

A *Miliolidium* sequence was obtained from isolate 16. The genus *Miliolidium* has been reported as a symbiont of Foraminifera and Porifera, or as free-living (*Pochon & LaJeunesse, 2021*). It is unclear if our isolate is novel compared to *Miliolidium leei*, the type species. While sequences in this clade are genetically divergent from each other, our data did not have an associated culture that could aid in differentiating these species (morphologically). Still, the diversity of hosts that acquire *Miliolidium* make this genus an interesting target for understanding symbiosis between microalgae, metazoans, and protists.

*Amphidinium* and *Heterocapsa* were also sequenced in this study. *Amphidinium* from isolate 61 (Fig. 8B) and those from Ikenoura (from 2019 and 2022) (Fig. 3) were identified as *A. gibbosum* based on morphology and molecular phylogenics (Fig. S1). *Amphidinium gibbosum* has been reported from *Waminoa litus* and *Heterochaerus langerhansi* (*Hikosaka-Katayama et al., 2012*; *Taylor, 1971*). Both *Waminoa* and *Heterochaerus* are relatively large acoels (measuring approximately 2 mm). As *Amphidinium* are much larger, in comparison to other acoel symbionts, they may occur more frequently in larger acoels. Future collections, targeting larger acoels, may uncover a higher diversity of *Amphidinium*. *Heterocapsa* species have never been reported as a symbiont of acoels. The single *Heterocapsa* sequence in our data likely represents a food, or other environmental contaminant, not a symbiont of acoels. *Heterocapsa* was neither abundant nor cultured in our experiments.

## Identity of acoels belonging to Convolutidae collected in this study

Fourty-four *Tetraselmis*-containing acoels were collected in this study. These are all believed to be Convolutidae, based on the presence of microalgae (*Atherton & Jondelius, 2022*; *Jondelius & Jondelius, 2020*). We observed two morphotypes, distinguished by the presence of colored rhabdoids (compare Figs. 2A to 2B). Of the 16 Convolutidae isolates sequenced (using 18S), individuals with colored rhabdoids formed a clade consisting of 14 identical sequences. This included isolate 33 which did not possess colored rhabdoids and contained *Symbiodinium* instead of *Tetraselmis*. This clade resembles the genus *Convoluta* —based on the distribution of colored rhabdoid gland cells, and body size (*Hooge & Tyler, 2008*). Only two individuals without colored rhabdoids (isolates 1 and 17) were sequenced. Both differed from any known *Symsagittifera* or *Convolutriloba* sequences; but they are believed to belong to one of these families as sagittocysts were visualized in a representative photo taken under oil-immersion (Fig. 2A). Isolates 1 and 17 (Figs. 1B and 1C) were only imaged with an inverted microscope, and sagittocysts were not apparent. Three of the isolates from Ikenoura (isolates 19–21), although lacking corresponding sequences, were identified as putative *Amphiscolops*, based on the three posterior caudal lobes and the presence of two symbionts (*Tetraselmis* and *Amphidinium*). This is similar to other *Amphiscolops* species.

Even though a large portion of the acoels collected in this study were not sequenced, it is likely that these represent a variety of lineages in Convolutidae. For example, light

micrographs of isolates 5 and 57 (Figs. 6K and 6L) have a sparse distribution of rhabdoids, a blunt posterior end, and a centrally-located statocyst (not anterior) (*Achatz, 2008*; *Achatz et al., 2010*). Similarly, isolates 58 and 59 (Figs. 4J and 4K) have notably rounded anterior and posterior ends. Molecular identification of acoels was challenging because host individuals were broken apart to establish microalgal cultures. This likely lowered the success rate (~70% of hosts were not sequenced). Breaking apart the hosts also made histological observations impossible.

## A potential novel family of acoels containing Symbiodiniaceae outside the Convolutidae

One of the more exciting discoveries from this study was a novel clade of acoels containing symbionts that fall outside the Convolutidae. The acoels from this clade contained a diversity of Symbiodiniaceae symbionts from *Symbiodinium* (subclades $A_{II}$, $A_{III}$, and $A_{IV}$) and *Miliolidium*. This is potentially the first report of non-convolutids containing microalgal symbionts. A prominent diagnostic characteristic of the family Convolutidae is the presence of microalgal symbionts—no other acoel family is known to contain microalgal symbionts (*Achatz et al., 2013*; *Jondelius & Jondelius, 2020*). This novel clade (seven individuals; isolates 2, 10, 12–16) was grouped sister to the Mecynostomidae, with only limited ML/BPP support (69/0.98). The shape and size of copulatory organs, or the presence of bursal nozzles could not be observed under bright field microscopy because the microalgal symbionts were dense. Most individuals also were not fully mature. Therefore, it was difficult to compare the morphology of these outstanding acoels to a contemporary taxonomic key (*Jondelius & Jondelius, 2020*). Histological slides and sperm axoneme structure are needed to confirm the identity (family) of this clade (*Achatz et al., 2010*).

## Microalgal symbionts are diverse, but what drives preference?

The acoel-microalgal symbiosis is likely driven by the needs of the host acoel. If juvenile acoels do not take up symbionts from the environment, they fail to fully mature (*Douglas, 1983*). Vertical transmission of microalgae from parent acoels to offspring is rare and is only known in *Waminoa* (*Hikosaka-Katayama et al., 2012*). Acoels provide nitrogen for microalgal symbionts, and in return the microalgae provide necessary photosynthates for the acoel (*Meyer, Provasoli & Meyer, 1979*). The mechanism behind the recognition and selection of symbionts is still unknown, however, the acoel *Symsagittifera roscoffensis* is selective towards *T. convolutae* (*Arboleda et al., 2018*).

Observations in the present study suggest that meiofaunal acoels prefer either *Tetraselmis* or Symbiodiniaceae. Acoels appear to be repeatedly associated (loyal) with one of these symbionts. Acoels with and without colored rhabdoids all hosted *Tetraselmis* symbionts. However, isolate 33, a convolutid hosting *Symbiodinium*, was also observed. Isolate 33 was genetically identical to 13 other convolutids that all harbored *Tetraselmis*. This means that a single acoel species can contain either *Tetraselmis* or Symbiodiniaceae; but this was not observed concurrently within the same individual.

There were subtle patterns that could be observed within our dataset at the species level. Isolates 5 and 57, which were morphologically different from other acoels, harbored

*Tetraselmis* belonging to the *T. marina* clade. Similar observations were made with isolates 58 and 59, with symbionts that were closely related to *T. convolutae*. *Arboleda et al. (2018)* demonstrated that *S. roscoffensis* could associate with six other species of *Tetraselmis*, albeit with varying levels of abundance. For example, *T. rubens* and *T. subcordiformis* appeared less dense within the host acoel, relative to *T. convolutae*. A larger sample size, based on molecular data from acoel hosts, would be needed to substantiate further claims regarding host-symbiont preference. In our study, dinoflagellates (*Amphidinium*) and *Tetraselmis* appeared together in only one acoel, the putative *Amphiscolops* from Ikenoura. *Tetraselmis* from these isolates were closely related to *S. roscoffensis* (*Parke & Manton, 1967*) and *Amphiscolops oni* (*Asai et al., 2022*). *Tetraselmis* found in these acoels seems to be limited to a single lineage, but this may have resulted from a sampling bias, as *Amphiscolop*-like acoels were not collected from Okinawa or Ishigaki. This association of *Amphidinium* and *Tetraselmis* might also be attributed to the relative size of the acoel (*e.g.*, *Amphiscolops*) (*Achatz, 2008*; *Asai et al., 2022*; *Ogunlana et al., 2005*; *Trench & Winsor, 1987*).

Regarding seasonality, no clear patterns were observed with respect to either *Tetraselmis* or Symbiodiniaceae. While seasonality cannot be completely disregarded as a driver of symbiotic preference, regular sampling at set locations targeting the same acoel species would have to be performed, to address such questions.

## CONCLUSIONS

Meiofaunal marine acoels in Japan host a variety of *Tetraselmis* and Symbiodiniaceae species. The majority of host acoels collected in this study were members of the family Convolutidae. We also introduce intriguing evidence that there may be a family outside of Convolutidae that harbors microalgal symbionts. Our survey of symbiotic microalgae found at least four lineages of *Tetraselmis* from acoels, including lineages closely related to *T. convolutae*, *T. astigmatica*, *T. marina*, and *T. gracilis*. This is the first report of *Tetraselmis* species, other than *T. convolutae*, as symbionts of acoels. Analysis of these sequences further highlights the need for taxonomic and systematic revision of *Tetraselmis*. Symbiodiniaceae from acoels included *Symbiodinium* lineages in subclades $A_{II}$, $A_{III}$, and $A_{IV}$, as well as *Miliolidium* and *Cladocopium*. Lineages from *Symbiodinium* and *Miliolidium* were identified for the first time as symbionts of acoels. We also were able to culture multiple lineages of Symbiodiniaceae from a single acoel. This shows that acoels containing multiple types of Symbiodiniaceae do exist, but their prevalence is unknown. Therefore, the true diversity of these acoel-dinoflagellate associations is likely still underestimated. Further genomic sequencing information would shed light on this issue. Strains of *Tetraselmis* and Symbiodiniaceae established in this study will be an important tool for taxonomic and experimental work. Future studies should focus on the identification of both acoels and symbionts using molecular data. This would be useful in mapping out the selectivity, seasonality, feeding ecology, and the symbiotic relationship between acoels and their microalgae.

## ACKNOWLEDGEMENTS

We would like to express our gratitude to the Takehara Marine Laboratory, Hiroshima University (Dr. Yusuke Kondo and Dr. Susumu Ohtsuka) for providing laboratory space during our sampling in Hiroshima/Kochi. We would also like to thank the members of the Wakeman Lab (Eric Odle and Dr. Samuel Kahng), Biodiversity Division (Dr. Hiroshi Kajihara) at Hokkaido University, Dr. Yonas Tekle at Spelman College, and Dr. Jonathan Banks for taking the time to read through drafts of the manuscript and/or provide constructive feedback throughout the course of this project.

### Funding

This work was supported by the Japanese Society for the Promotion of Science (No. 18K14774). The authors were also supported by the ISP program at Hokkaido University. There was no additional external funding received for this study. The funders had no role in study design, data collection and analysis, decision to publish, or preparation of the manuscript.

### Grant Disclosures

The following grant information was disclosed by the authors:
Japanese Society for the Promotion of Science: 18K14774.
ISP program at Hokkaido University.

### Competing Interests

The authors declare that there are no competing interests.

### Author Contributions

- Siratee Riewluang conceived and designed the experiments, performed the experiments, analyzed the data, prepared figures and/or tables, authored or reviewed drafts of the article, and approved the final draft.
- Kevin C. Wakeman conceived and designed the experiments, performed the experiments, analyzed the data, prepared figures and/or tables, authored or reviewed drafts of the article, and approved the final draft.

### Field Study Permissions

The following information was supplied relating to field study approvals (*i.e.*, approving body and any reference numbers):

Hokkaido University; No field permits were required for this study.

### DNA Deposition

The following information was supplied regarding the deposition of DNA sequences:

The ribosomal sequences generated in this study are available at NCBI GenBank: OQ439302; OQ439515–OQ439516; OQ449206–OQ449249; OQ449250–OQ449284; OQ779018–OQ779040.

## Data Availability

The sequence data, sample data, and the primer data are available in the Supplemental Files.

## Supplemental Information

Supplemental information for this article can be found online at http://dx.doi.org/10.7717/peerj.16078#supplemental-information.

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
