# Peer review of "Biodiversity of symbiotic microalgae associated with meiofaunal marine acoels in Southern Japan"

_PeerJ, doi:10.7717/peerj.16078_

## Round 0.1 · original submission · Minor Revisions

I have received evaluations of your manuscript and these can be seen below. The three reviewers have made extensive comments on your manuscript and are in agreement that a minor revision is required. Please ensure that you attend to all of the suggestions in a revised version of the manuscript.

Reviewer 1 ·

Basic reporting

The language is clear and professional. References are sufficient. The manuscript followed the journal structure, and is self contained. Supplementary material is all accessible and includes raw sequence data.
The Introduction is a bit short. More information on the Convolutidae group would help to put into context the importance of this work. As is written, the introduction only comments on the diversity and symbiotic nature of Convolutidae acoels.

Experimental design

The primary question is clearly identified and relevant. The knowledge gap for the biodiversity of symbionts associated with Acoel invertebrates is stated. However, the Ms would benefit if more information on this Convolutidae group is given, such as its ecological importance, if any (as I pointed for the introduction).
Methods are described with sufficient detail.
Authors report on the identification of symbionts, but in the Results no attention is given to the Amphidinium genus. Perhaps the authors grouped Amphidinium with Symbiodiniaceae?

Validity of the findings

Sequence data deposited, accession provided.
Conclusions are appropriate to the intended research and are supported by the results obtained.

Additional comments

I consider figures 4 to 8 too large. There are only a few individual photographs that are addressed in the discussion. I suggest moving some of the photographs to supplementary material to make these figures more condensed.

Annotated reviews are not available for download in order to protect the identity of reviewers who chose to remain anonymous.

Reviewer 2 ·

Basic reporting

No issues.

Experimental design

No issues.

Validity of the findings

No issues.

Additional comments

Riewluang and Wakeman have carried out a useful and interesting survey of symbiotic microalgae inhabiting acoels in Japan. The methods are appropriate and the conclusions are sound. I have no suggestions for changing the analyses, just suggestions to improve clarity. The manuscript has the feel of an early draft.

The introduction was lacking in detail and specifics. The background information about acoel biology and diversity was not sufficient to give a broad audience the necessary foundation for understanding the paper. Several statements in were vague, with a tendency to mention the general topic of a previous study rather than the actual results. The introduction should be expanded and revised for detail and coherence.

Some important details were missing from the results section too, notably identification of the host organisms. The discussion continued in a similar tone to the results with lists of data rather than interpretation and contextualization of the results. These sections should also be revised, with clearer explanation of the state of knowledge before this study and how the new results from this study have advanced or altered our knowledge of acoel symbionts. Specific suggestions to address these general points follow:

Abstract
line 22: change to “symbiotic microalgae” or “a symbiotic microalga”

Introduction
line 35: include the taxon level of “Acoelomorpha” and include a higher taxon for orientation (e.g., Metazoa) and/or at least a common term like “invertebrate animal”.
line 36: More information about the body plan of acoels would be useful here. Please define the anatomical terms used later in the article, such as planula, rhabdoid cells, statocysts, eyespots, saggitocysts, and whatever features distinguish convolutids from other families. Which lineages are model organisms? How diverse is the group? How many genera? Species? What characteristics are used to define genera? Is anything known about their ecology or behavior? A paragraph or two should be sufficient, as long as aspects of relevance to the rest of the paper are covered.
line 40: please disambiguate “these species”: does it refer to the acoels or the microalgae?, e.g. Most Convolutidae species …
line 52: More specifics, please. What have the “few works” reported?
line 55: Include higher taxa to orient the reader of the systematic position of Tetraselmis.
line 57: “cultures of green algae” presumably refers to one or more species of Tetraselmis? If so, specify which one(s). If not, this is not a relevant statement for this introduction.
line 60: replace “ambiguous” with a more appropriate descriptor, maybe “inadequate” or “understudied” or “unresolved”.
lines 61-62: Please be more specific about the defining features of Tetraselmis and what can vary about their appearance.
line 63: “The phylogeny of this group has been questioned” doesn’t quite make sense. One specific phylogenetic hypothesis may have been questioned, but certainly not the actual phylogeny of the group. Please be more specific. Presumably it is a traditional morphology-based evolutionary scenario that has been questioned? Or the taxonomic utility of certain morphological characteristics has been questioned?
line 66: “identified to the species level” Would it be more accurate to say that only one of the symbiotic Tetraselmis species from convolutids has been named and described? This sentence is a bit confusing because the beginning of the paragraph said that all Tetraselmis species can be found free-living yet this species seems to be named after a host lineage. Perhaps also indicate how well the 30 described species encompass the morphological and molecular diversity of the genus.
line 67: Please reword this last sentence to make the point clear. If acoels can acquire closely related Tetraselmis species, the diversity across acoels could just as easily be lower than expected.
line 76: What type of cytological effects? What do we expect amphidiniols to do?
line 80: Which species of Waminoa?
line 88: Perhaps some additional justification could be provided for the study? Investigating biodiversity is a worthy goal, in my opinion, but the motivation of the study would be clearer if there were a more forward-thinking rationale. Why do we want to know about the diversity of acoel algal symbionts? To establish symbiont/host model systems? To compare with other photosymbioses such as corals or protists? To determine the host breadth of symbiotic microalgal species? There are many possibilities.

Methods
lines 95-96: Please provide more information about sample collection. The introduction suggests that some acoels live as epiphytes on macroalgae while some live in sand. What exactly was sampled in these intertidal and subtidal zones? Why were these locations chosen?
line 97: Should this read “acoels were anesthetized”? Or maybe “magnesium chloride was added to the samples to anesthetize any animals that may be present”? Again, this might make more sense if the samples had been described more fully.
line 98: Why this mesh? Was this to include or exclude the acoels? Presumably to include while excluding smaller debris, but it helps to include statements like “in order to …” before describing a method.
line 109: change to “Daigo’s IMK medium”
line 111: include a space between 25 and ºC
line 112: change “visible” to “visibly pigmented”
line 118: change to “was extracted” or “Genomic DNA samples … were extracted”
line 130: “apart from general eukaryotic primers” which are these? They haven’t been mentioned yet.
line 134: What is the nested reaction? So far only single PCR reactions have been described.
line 145: change to lower case s on Sequencing

Results
lines 172-178: Are these morphological characteristics sufficient to identify the acoels to genus level? Supplementary Figure 1 indicates “Acoela” as the genus for some isolates, how was this identification made?
lines 172-189: This section should have a lot more information about the morphological and molecular identification of hosts, including how many had molecular data and which ones could be identified to genus morphologically and why.
line 184: Did these four genetic groups correspond to the four morphotypes or not?
line 195: Fig. S2 is a phylogeny of Amphidinium. I do not understand why this is referenced here after the statement that isolates 19-21 have multiple symbiont types.
line 196: How were cultures established for both isolates? Serial dilution? Or are they still in co-culture? This should be mentioned in the methods section.
line 205: What evolutionary model was used to “correct” the pairwise distances? This should also be described in the methods section. Or perhaps this should say “uncorrected”?
line 215: Maintaining parallel structure would make this section easier to follow. Rephrase this sentence to say “The Miliolidium clade included isolate 16 from Kabira …” and then in the next sentence, “The Cladocopium clade included…” etc.
line 225: Please clarify what these genetic distances represent. Intra-clade diversity? Distances between these last listed sequences and their sister clade?
line 229: Rather than “sequenced”, perhaps say “identified as symbionts of acoels”
line 229: “A culture” from “Convolutidae isolates”? Please clarify - presumably the culture only came from one isolate?
line 233: Replace “physically” with “directly”

Discussion
lines 237-270: This information is more results than discussion. Please discuss the correspondence of Tetraselmis phylotypes with host types.
line 238: Please be more specific about the results of previous studies. It is not informative to say “this is the first time such a diversity of Tetraselmis has been documented in acoels”
line 297: “largely” or completely based on morphological information? be specific about the results of previous studies
lines 303-326: This section also belongs in results more than in discussion.
lines 341-343: Were sequences acquired for these putative Amphiscolops? Please specify. These samples don’t seem to appear in the host phylogeny fig. S1.
line 345: It would be helpful to state more clearly and earlier in the manuscript that molecular data weren’t acquired for many of the acoels. Add this information to the “sampling and identification of host acoels” section in the results.
line 350: Insert “molecular” before “Identification”
line 362-363: Please clarify “it was difficult to compare these outstanding sequences” to what? To each other? To the morphology?

Figures
Fig. 1: I’m a bit confused by the labeling “Japanese Archipelago” below the inset of Shikoku and “Southern Mainland Japan” closer to the Ryukyus. Consider removing these labels? I don’t think they're necessary.
Fig. 2: Beautiful images! Check terminology for consistency re: ocelli/eyespots. The legend says Ocelli (Oc) but these are indicated with “E” on panel A, and are referred to as “eyespots” in other figures. Also please indicate the sample number and collection location for each of these images in the legend.
Fig. 3: Again check eyespot labelling, here the legend says “eyespot (E)” but the image says “Oc”.
Fig. 4: The isolate number was only given for panels A, B, J, and K. What were the others? Please provide some additional interpretation for the rest of the panels, like what is shown by these images that can’t be seen in A, B, J, or K? Why are all these images here?
Fig. 5: Again, only G and L are identified specifically. Why are the other panels included?
Figs. 6 to 8, same comment. The images are beautiful, but it is not clear why they are included or why so many need to be shown. Perhaps the images of specific samples from Figures 4-8 should be combined into a single figure and the rest put into the supplemental? Alternatively, provide more justification for including so many panels in each figure.

Supplementary Material
The table of primers is listed in the manuscript as Table S2, but the word document says “Supplementary Table 1” at the top of the table.
Please provide figure legends for the supplementary figures, especially Fig. S3.

Reviewer 3 ·

Basic reporting

No comment.

Experimental design

This is very minor, but I would have liked more detail on the microalgal culturing techniques.

Validity of the findings

No comment.

Additional comments

The authors investigated the diversity of symbiotic microalgae that associate with meiofaunal acoel worms found around Japan. Using microscopy and ribosomal DNA sequencing, they identified green algae (Tetraselmis) and dinoflagellates (Amphidinium and members of the Symbiodiniaceae) as the primary symbionts, with rare incidents of multiple symbionts residing within one host animal. Acoel symbionts have rarely been examined outside of a classic model system, so this study provides some much needed ecological and biogeographic insights into the diversity of such associations in nature.

This is a straightforward study (in a good way), with clear goals, methods, and analyses, along with conclusions that flow logically from the results. I’m interested in dinoflagellates, so I was particularly excited about the discovery of Symbiodinium natans and Miliolidium associating with acoels. It might have been useful to provide a bit more context for some of the morphological characteristics used to distinguish acoels (I had to look up rhabdoid gland cells, for instance), and I would have preferred more detail regarding microalgal culturing techniques. Some figures could use more descriptive captions, or perhaps fewer images. But these suggestions—along with those that follow—are all minor issues that should be easily addressed in a quick revision. Overall, I feel the study is rigorous and entirely suitable for PeerJ.

Minor Comments

L82-85: Minor clarifications: “Nevertheless, A. gibbosum is the only acoel-associated dinoflagellate that has been identified to the species level (Taylor, 1971; Trench & Winsor, 1987), and Cladocopium is the only acoel-associated Symbiodiniaceae genus that has been identified with genetic data (Kunihiro & Reimer, 2018).”

L108: Could use more detail about microalgal culture establishment. What volumes were used? How were cells picked? Were they started from single cells or multiple? What was the transfer frequency? What was the light intensity?

L110: I would state explicitly that the reason host + symbiont material from acoels that were used to established microalgal cultures were preserved for DNA extraction was to confirm that the subsequent culture represented the natural symbiont (and wasn’t a contaminant).

L123: Can you explain why you used a nested PCR approach?

L200: When you say cultures were established and sequenced, it’s a bit vague as to whether the sequences from isolates matched those from derived cultures. For the Symbiodiniaceae, at least, it doesn’t appear based on the phylogeny than any successfully sequenced isolates yielded cultures, which is a detail worth noting. The success rate for Tetraselmis seems higher.

L223: Typo: it should be S. tridacnidorum (with an ‘o’). Please check throughout the manuscript.

L314: Symbiodinium natans is not (yet) known as the primary symbiont of any hosts (acoels, corals, or otherwise), but it has been cultured multiple times from coral isolates. In these organisms it’s either a background symbiont or a free-living contaminant. It would be very cool if it turns out acoels are the true host (and a potential source of culture contamination when isolating Symbiodiniaceae from corals), and I would suggest pointing this out in the manuscript. The most logical place would be in the discussion L314. The story of how S. natans is not a known symbiont can be found in the supplement of LaJeunesse et al. (2018) Current Biology in the S. natans species summary.

L365: Typo: “identity”

Figure 7: The caption should highlight acoel B, which hosts both Symbiodiniaceae and Amphidinium (according to L182). Why are the other images included if they are not described? Is it just to show the morphological diversity?

---

## Round 0.2 · Minor Revisions

I am satisfied with the changes that have been made to the manuscript. The only excpetion is that the materials and methods section should be written in the past tense but it varies between present and past tense. With these minor corrections I will be more than happy to propose that this manuscript be accepted for publication.

---

## Round 0.3 · accepted · Accept

Thank you for editing the manuscript. It now reads well.